# Quantum Randomness is Chimeric

**DOI:** 10.3390/e23050519

**Published:** 2021-04-24

**Authors:** Karl Svozil

**Affiliations:** Institute for Theoretical Physics, TU Wien, Wiedner Hauptstrasse 8-10/136, 1040 Vienna, Austria; svozil@tuwien.ac.at

**Keywords:** quantum randomness, Gleason theorem, Kochen–Specker theorem, Born rule, object construction, emergent space-time, quantum entanglement, 03.65.Ca, 02.50.-r, 02.10.-v, 03.65.Aa, 03.67.Ac, 03.65.Ud

## Abstract

If quantum mechanics is taken for granted, the randomness derived from it may be vacuous or even delusional, yet sufficient for many practical purposes. “Random” quantum events are intimately related to the emergence of both space-time as well as the identification of physical properties through which so-called objects are aggregated. We also present a brief review of the metaphysics of indeterminism.

## 1. Quantum Oracles for Randomness

Almost 40 years since the glamorous inception [1,2,3,4,5] of quantum computing, and despite numerous grandiose claims and prospects of quantum computational advantages [6,7,8], only the random generation of bit sequences by beam splitters [9,10,11,12,13,14,15,16,17] has reached a certain commercial [18] maturity. Yet, these quantum random number generators present oracles [9,19] for “randomness”, which (i) inductively are imagined and extrapolated to be a finitistic version of an essentially transfinite concept [20]. “Certifications” by NIST and DIEHARD and other sophisticated test suites are of little consolation; and other natural resources for randomness exhibit similar performances [14,17]; and (ii) deductively are certifiable merely relative to the principles, assumptions, and axioms—such as, for instance, complementarity or “contextuality” [12,16,21,22]—they are based upon. It is therefore of utmost importance to carefully delineate and be aware of these latter presumptions if we want to certify and trust such devices.

In what follows, we shall discuss randomness “extracted” from measurements of coherent superpositions of classically mutually exclusive states, then proceed to multipartite and mixed states. No quantum field theoretic many-particle effects such as stimulated or spontaneous emission or decay will be mentioned. In the later parts of the paper, we shall attempt a brief history of physical events that have been deemed “random” and, in particular, their relationship to the metaphysical ideas implied.

I encourage the reader to consider some of the content speculative and challenging—not as disrespectful to proposals and operationalizations of quantum randomness, including some earlier ones I myself contributed to [9,12,15,23]—but as reflections on some aspects that might be noteworthy and even troubling. A recent “canonical” presentation of quantum randomness in a broad perspective can be found in Reference [24]. One might also add that certain interpretations of Everett’s relative state formulation [25] suggest similar conclusions, albeit for very different reasons: that randomness is an intrinsic illusion [26].

### 1.1. Quantum Randomness through the Measurement Problem

Quantum mechanics allows the coherent superposition (or, by another denomination, linear combination) of states which correspond to mutually exclusive outcomes. The question arises: what kind of physical meaning can be given to these “self-contradictory” states? Furthermore, is it not amazing that, for such states, there exist two types of very closely related measurements that give vastly different results: one random and one not?

Let me explain this question in some more detail for an ideal configuration, thereby neglecting observational (or measurement) errors; in particular, no stochastic or random errors are taken into account. Suppose one prepares a pure quantum state, say, by pre-selecting certain outcomes of beam splitter experiments. (By similar arguments as the ones exposed here the randomness of mixed states are epistemic rather than ontic, and therefore, for all practical purposes, chimeric as well.) If one “measures” these pre-selected states again and again by serial composition of either identical beam splitters, or “contextual intertwined” beam splitters, one of whose output ports “shares” (and corresponds to) the pre-selected state [27,28,29,30], then a detector registering (or post-selecting) the “resulting” states (“after such serial processing”) will always click with certainty. That is to say, such an experiment reveals a strictly deterministic, absolutely predictable behavior of this pre-selected quantum state.

Even the slightest physical “tilt” or “rotation” of one of the serially composed beam splitters changes the situation entirely and dramatically: according to the standard quantum narrative, the experiment suddenly and discontinuously “performs indeterministically”, such that individual events—or at least post-processed sequences of such individual events—turn out to be irreducibly random [31] (relative to maybe “mild” side assumptions, such as independence, any bias can be eliminated by (Borel) normalization [32,33,34]). Such a physical manipulation of the beam splitter—literally “tilting” or “rotating” it—translates into a unitary transformation; that is, a generalized “rotation”, of the state (or context) or (by the dyadic products) the respective observable proposition(s) in Hilbert space. A “slight detuning” associated with a “small” change of the post-selected context with respect to the pre-selected context will not “throw the outcomes into crazy randomness”. Indeed, the quantum probability is a smooth function of detuning, so a “slight detuning” will only introduce a “small” amount of indeterminism in the raw data extracted. Nevertheles, relative to certain mild side assumptions such as independence of events, any such “tiny signal” of indeterminism in the raw data can be “amplified into crazy randomness” by (Borel) normalization, such as von Neumann’s [32] partitioning of the raw data sequence into subsequences of length two, and then mapping 00↦∅, 11↦∅, 01↦0, and 10↦1. This sudden, discontinuous change from determinism into complete indeterminism by some “smooth, continuous” manipulation boggles a mind prepared to “evangelically” [35,36] accept the quantum canon.

For the sake of a concrete example, take $|\psi \rangle =\psi_{0}|0\rangle +\psi_{1}|1\rangle ={\left(\begin{array}{c} \psi_{0},\psi_{1} \end{array}\right)}^{⊺}$ with $|\psi_{0}{|}^{2}+{\left|\psi_{1}\right|}^{2}=1$ and (⊺ stands for transposition), $|0\rangle ={\left(\begin{array}{c} 1,0 \end{array}\right)}^{⊺}$ and $|1\rangle ={\left(\begin{array}{c} 0,1 \end{array}\right)}^{⊺}$. Suppose we prepare or pre-select the quantized system to be in the state $|\psi \rangle =\frac{1}{\sqrt{2}}\left(|0\rangle +|1\rangle \right)=\frac{1}{\sqrt{2}}{\left(\begin{array}{c} 1,1 \end{array}\right)}^{⊺}$, and we prefer to measure an observable |ψ〉〈ψ| (that appears “rotated” or transformed relative to the observables |0〉〈0| and |1〉〈1|). In such a case, the system presents itself as being perfectly determined and value definite; with the respective outcome always occurring. No randomness or value definiteness can be ascribed to such a configuration. (Value definiteness shall be understood as “possessing” a well-defined property, encodable by some mathematical entity. In terms of (ideal) measurements, value definite properties yield the respective outcomes with certainty.) With respect to |ψ〉〈ψ| and its perpendicular orthogonal projection operator $1_{2}-\left|\psi \rangle \langle \psi \right|$ there is no uncertainty, and no possibility to obtain randomness.

Randomness comes about if “detuned experiments” are performed, such as, for instance, the ones “measuring observables” corresponding to the orthogonal projection operators |0〉〈0| and $\left|1\rangle \langle 1\right|=1_{2}-\left|0\rangle \langle 0\right|$. This concrete example features maximally or mutually unbiased [37] bases; but any “tiny” rotation 0≠φ≪1, with $\psi_{0}=\cos\varphi$ and $\psi_{1}=\sin\varphi$ suffices to yield irreducible randomness through (Borel) normalization, as mentioned earlier.

An immediate question arises: why should such “tilted” or “detuned” experiments yield any results at all, and if so, in what way do outcomes of such “wrong experiments” come about; and to what extent do they reflect any intrinsic property of the pre-selected state |ψ〉? It is rather mind-boggling that one should get any answer at all from such queries or “detuned” measurements. However, this may be as confounding as it may be deceptive: because one might get the impression that there is a physical property “out there”, “sticking” and being associated with the state. I believe that mistakenly interpreting an experimental outcome—such as a detector click—as some inherent property, constitutes a major epistemological issue that underlies many ill-posed claims and confusions about such quantum states. Indeed, these misconceptions may epitomize erroneous claims upon which quantum number generators by “quantum coin tosses” are based.

The quantum measurement problem is relevant for any judgment or certification or opinion on quantum randomness: “extracting” or “reducing” such states as |ψ〉 by “measuring” them in the “wrong and detuned” basis |0〉 and |1〉, different from |ψ〉 and its orthogonal vector, lies at the heart of the quantum measurement problem. The respective “process”, just as taking (partial) traces, is non-unitary because it is postulated “many-to-one” and irreversible. Therefore, such “processes” are inconsistent with the unitary quantum evolution, which is “one-to-one” and reversible. (see Section 1.8 of Ref. [5] for a nice presentation.)

This inconsistency is an old issue that has already been raised by von Neumann [38,39], Schrödinger [40,41,42], London and Bauer [43,44], Everett [25,45,46], and Wigner [47]. It can be developed as a “nesting” or “inverse Russian doll” type argument by ever-increasing the domain of unitarity; including the measurement apparatus and the measured state, and hence the interface or cut “between” them. This has been proposed and operationalized in quantum optical experiments reconstructing the coherent superposition of states after “measurements” [48,49,50,51,52,53,54,55,56], as well as in discussions about the insurmountable practical difficulties in doing so [57,58].

Strictly speaking, by assuming irreversible many-to-one “processes”, one has to go beyond quantum mechanics in an *ad hoc* fashion. Presently, there is no evidence suggesting that this is necessary or even consistent with empirical data. Should quantum mechanics be extended against all experimental evidence, just because it is theoretically convenient and saves primitive notions of “measurement”?

### 1.2. Objectification by Emergent Context Translation

In what follows, it will be argued that any kind of measurement—in particular, also associated with “detuned experiments”—constitutes an object or reality construction, whereby the conventionality of measurement plays an essential role. In this process, the very notion of objects or physical properties becomes conventionalized. Objects or the properties constituting them may be real or chimeric; in the latter, chimeric case those experiments relate to properties the system is fantasized about, but not encoded in [59]. In a metaphorical sense, this is like map-making or the creation of an encyclopedia, in which entries are constituted as facts or fiction, or in any other way that is supposed to be consensical or intentional.

The term “chimeric” will be associated with coherent superpositions or linear combinations of different (mutually orthogonal) states, relative to those states or their associated observable propositions involved. For instance, $|\psi \rangle =\psi_{0}|0\rangle +\psi_{1}|1\rangle$ with nonzero $\psi_{1}$ and $\psi_{2}$ is chimeric relative to the propositions |0〉〈0| and |1〉〈1|; but is value definite, or “real”, and not chimeric relative to |ψ〉〈ψ|. States are not chimeric relative to the propositions associated with those exact states, that is, |ψ〉 is “real” and not chimeric relative to |ψ〉〈ψ|.

The emergent process of “creating chimeras” will be called “objectification” or object emergence or (re)construction. Objectification is related to an ancient conundrum [60]: the *Ship of Theseus*, or more generally, what is in Philosophy called “the problem of identity” [61,62]. In the physical measurement process, it is the question of how, through “mediation” of its environment and the measurement apparatus, a physical state or system which initially is unprepared to answer a particular query—or, stated differently, is value indefinite and chimeric—“translates” the respective “detuned” query such that it is can respond to the request. Through this “context translation”, it may have acquired signals and information exterior to itself, which may render the answer stochastic relative to itself (because of an influx from the open environment) and to the experimental means available [63,64] (containing or severing that open environment).

One might object that this stance reiterates a well-known fact: that quantum measurement introduces stochasticity. The point of departure from this common view is the emphasis on the “nesting” aspect of the situation, as outlined already by Everett [25] and Wigner [47]; but unlike them, more in the spirit of statistical physics: In a Maxwellian view [65], the stochastic behavior (and entropy increase) originates from sampling—from not looking at the micro-physical level, but at some “aggregates”—rather than taking this for granted.

This has consequences for the stochasticity of chimeras: they are not only based on some property intrinsic to the object, but on the combined context by which the object, as well as the apparatus, is defined [66]. Stochasticity enters by the many degrees of freedom of such a combined system. This kind of emergence of an “experimental outcome” associated with a counter reading of a (macroscopic) measurement apparatus has already been modeled (i) by a coupling of the object with the apparatus and its environment [67], and (ii) by “attenuating” a quantum signal from a state to cloning a “noisy multitude” of this state [68,69] (it is always possible to clone two fixed orthogonal states) “as much as possible” (that is, nothing at all) within the framework of the no-cloning theorem (cf. Section 2.1 of Ref. [5]).

For the sake of understanding on which basis claims of absolute randomness are raised beyond evangelical confessions [31,36], let me reconstruct current “best-practice arguments” for quantum indeterminacy [70] and value indefiniteness [22,71] and their counterfactual [72,73] character. There “exist” collections of (counterfactual [72]) observables comprising intertwining contexts (formalized by orthonormal bases or maximal operators in dimension three or higher) with two terminal point states—one serving as pre-selection or preparation, the other one for postselection or “measurement”—with the following inconsistent properties: Upon pre-selection or preparation of a particular state |Ψ〉, (i) one such collection of observables enforces the nonoccurrence of some post-selected state |Φ〉, associated with a certain negative experimental result; (ii) another one such collection of observables enforces the occurrence of some post-selected state $|\Phi^{\prime }\rangle$, associated with a certain positive experimental result [74,75]; (iii) both post- and pre-selected states are the same, say, $|\Psi \rangle ={\left(\begin{array}{c} 1,0,0 \end{array}\right)}^{⊺}$ and $\left|\Phi \rangle =\right|\Phi^{\prime }\rangle =\left(1/2\right){\left(\begin{array}{c} \sqrt{2},1,1 \end{array}\right)}^{⊺}$ [22,71,76]. Figure 1 sketches such a configuration. The classical inconsistency arises from the fact that, depending on the arrangement of the quantum observables, the same observable must either be false (snake-like decorated curve) and true (zigzag-like decorated curve) at the same time—a complete contradiction amounting to the absurd prediction that a detector associated with such a binary observable simultaneously registers a click and does not do so. Relative to the assumptions made |Φ〉 given |Ψ〉 cannot have a classical value definite truth assignment: any such truth assignment would need to be undefined at least for |Φ〉. This yields the truth assignment as a partial function, a notion well known in theoretical computer science [77] The argument can be extended to any state not collinear with or orthogonal to the pre-selected state |Ψ〉 [22].

Another implicit assumption that is seldom mentioned because it is assumed evident is the omni-existence of the collection of complementary observables (because the argument involves different contexts). Indeed, the coexistence of counterfactual, complementary observables is (mostly implicitly) assumed without further discussion. One common response to critical doubts about their existence is that “they can be measured”. That is, a particular state |ψ〉 can be prepared or pre-selected and subsequently, the proposition corresponding to another “mismatching” state |φ〉 (which should neither be orthogonal to, nor collinear with, |ψ〉) can be measured or post-selected. This, of course, is omni-realism, pure and simple.

Coming back to the argument sketched in Figure 1, it is evident that, due to pre-selection or preparation, the state |Ψ〉 and its associated observable proposition |Ψ〉〈Ψ| is value definite relative to measurements |Ψ〉〈Ψ|. However, should this be assumed for all the other observables entering the argument? In particular, should value definiteness be expected from some state |Φ〉 given |Ψ〉? Because |Φ〉, and all other observables, entering as counterfactual “intermediaries” in the argument, need to be in a coherent superposition of states different from the pre-selected state |Ψ〉 and other states, which makes them chimeric relative to |Ψ〉.

### 1.3. Information Theoretic Approach to Quantum Randomness

A related information-theoretic argument for “irreducible” [31] quantum randomness contends that a quantum system can “carry” only a finite amount of information [59,78,79]—namely (maximally) about the occurrence of a single proposition within a single context. Therefore, the [59] “*…* reason for the irreducible randomness in quantum measurement *…* is the simple fact that an elementary system cannot carry enough information to provide definite answers to all questions that could be asked experimentally”. Stated differently [80], “there are less available answers than possible questions”. Any query attempting to forcefully retreive “more” information from such a quantized system is confronted with this “underdetermination”, resulting in ontological indeterminism.

Alternatively, one might argue that this “insistance on the enforced retrieval of information the quantum system is unprepared to hold” results in a context translation. Typical examples are “detuned experiments” mentioned earlier, associated with an influx of information from the environment and, in particular, the measurement apparatus. This effectively results in epistemic quantum indeterminism.

One could still maintain that, through nesting [25,47] and the effects of the translational environment the number of degrees of freedom during the measurement cannot be bounded from above and approaches infinity, resulting in nonseparable hyper-Hilbert spaces [81], a situation which might yield a sort of irreducible randomness based on the diverging complexity of the environment [82]. Note that even classically the hypothetical invocation of infinity “in the limit produces” provable random sequences, such as Chaitin’s halting probability Omega [83].

### 1.4. Entanglement and Emergence of Space-Time

Einstein’s primary intent [84,85,86] in writing a paper with Podolsky and Rosen [87] (EPR) was to present a separation principle or separation hypothesis: given any two (space-like) separated subsystems *A* and *B* of a joint system (AB), then *B* (my translation, see also [86]) “and everything related to its content is independent of what happens with regard to” *A*. Thereby, Einstein’s presumption has been that, after any interaction between *A* and *B* in the past (quoted from the same letter, my translation, see also [86]) “the real state of (AB) consists of the real state of *A* and the real state of *B*, which two states have nothing to do with one another”.

This latter assumption, at least for Einstein, is one pillar of the EPR argument. However, suppose that we are not inclined to follow Einstein’s critique of quantum mechanics, but propose that, rather than quantum theory, space-time physics, and relativity theory would need to adapt in case there is a collision with quantum mechanics. Then the separation principle should be considered incorrect and not be applied for entangled quantum states introduced by Schrödinger [40,41,88,89] around the time of the EPR paper. In particular, there exist entangled states of two subsystems *A* and *B* which are indecomposable; that is, they cannot be written as the product of the states of the two “separated” systems *A* and *B*; more formally, |Ψ(AB)〉≠|ψ(A)〉⊗|ϕ(B)〉, where ⊗ stands for the tensor product.

This inseparability, as discussed by Schrödinger in the measurement context (between object and measurement apparatus) has been re-interpreted in terms of relational properties [59] for multi-partite configurations. It comprises two parts—a restrictive and an extensive property for classical physical systems: (i) quantum mechanics limits the amount of information encodable in a quantized system from above; and (ii) it allows the storage, resampling [90], or scrambling of such limited information “across quanta”. Both properties can be viewed as direct consequences of the unitary transformations postulated as formalizations of quantum state evolution, because entangled systems are merely “a unitary transformation apart” from separable states ([91], Section 12.8.2).

Let us pursue a very radical, iconoclastic deviation from the Kantian idea that space-time is an *a priori* theatric frame, a sort of scaffolding, in which physics takes place. Rather, suppose that

(i)in reversing Einstein’s verdict mentioned earlier, for (maximally) entangled states of a composite system (AB), its constituents share a common identity—that is, they “are tied together” and can be considered “being aspects of a single entity” and, in particular, “not spatio-temporally separated at all”; so much so that any individuality or separateness vanishes.(ii)Space-time needs to be derived from quantum effects as an (emergent) epiphenomenon, a secondary effect or byproduct that arises vis-à-vis quantized systems and does not stand separate from or independent of them.

In this view, distances are a matter of disentanglement and gradual: two events such as detector clicks are “apart” if their corresponding states are (for all practical purposes) factorizable and decomposable, and thus disentangled. Spacio-temporal separations and distances are to be understood more like the second law of thermodynamics [65]: they are not absolute, but relative to the (entanglement) means involved. This creates a “patchwork” of clocks and rulers, associated with the respective entanglements. Such emergent space-time frames need not necessarily be consistent with one another, but rather form a mesh of spatial-temporal networks.

Most radically, what may be considered “far apart” in the old Kantian–Einsteinian framework maybe not be separated at all in the new scheme. For most practical purposes [92,93], the two notions of spatial–temporal distances may coincide. Because entanglement and “nonlocality” with respect to the old “absolute” theatrical framework of space-time (for all practical purposes) “happens locally” and—again according to the *Ancien Régime* in terms of Kantian–Einsteinian space-time frames—not “far away”.

This radical departure from the Kant–Einsteinian framework of space-time by emergence from entanglement has been discussed in entanglement-induced gravity [94,95,96,97,98,99,100,101]. See also Ref. [102] for another approach to emergent space-time. This research program is a new and active area of research.

A lot of questions arise immediately. One issue that needs to be addressed is that of the finite speed of light, as compared to instantaneous entanglement: can some finite speed of information transfer be derived from an infinite property? One Ansatz is given in Ref. [103]. What is (inertial) motion, and the type of kinematics resulting from entanglement? Entanglement swapping comes to mind immediately, but this lacks any notion of inertia. Indeed, we might be tempted to speculate that the absence of inertia, rather than being a problematic feature, might be an advantage, suggesting possibilities of inertialess motion [104], and motion beyond the relativistic speed limit. It might not appear too unreasonable to speculate, that, if entanglement swapping takes place instantaneously, so maybe motion or signaling in space and time, even despite the following discussion.

### 1.5. Peaceful Coexistence

The argument stated by Einstein in his letter [84,85,86] to Schrödinger quoted earlier amounts to the aforementioned separation principle: measurement of a subsystem *A* of (AB) cannot alter the state of the subsystem *B*; in particular, not if the two subsystems are spatially separated. As noted earlier, Einstein attacked quantum mechanics for failing this principle for entangled multi-partite states. However, as our approach considers the emergence of space-time as secondary to quantization, rather than questioning the validity of quantum mechanics, we might as well respond with an “upside-down” question: why not? Why is space-time not challenged by these issues? To answer such questions, it might be prudent to compare a similar classical EPR-type configuration with classical and more general resources. We can imagine at least two scenarios:(i)Value definiteness of the individual constituents *A* and *B* and the fixing of their respective local shares at creation point: for this scenario, Peres gave a most insightful analysis [105]. Classical “singlet” states (e.g., obtained by the preservation of angular momentum) may exhibit certain (dis-)similar behaviors as compared to the quantum case. Classically, the joint system (AB) “carries” some “common share”—e.g., a hidden parameter such as the opposite angular momentum pseudovectors of the particles [106,107,108] along one and the same direction. These angular momentum pseudovectors are fixed and value definite for both parties or subsystems *A* and *B* already after their interaction. Therefore, the local information can in principle be used to produce local “copies” or “clones” of *A* and *B*. This is consistent with relativity theory because those shares remain fixed after their creation, so that whatever manipulation happens on one side does not alter the respective state or share on the other side.(ii)Value indefiniteness of the individual constituents *A* and *B*, but the fixing of their respective global shares at creation point: This may for instance be achieved by assuming a global value definite share or state of (AB); and yet by not allowing or “granting” definite states to the individual constituents *A* and *B*. Therefore, any attempt to copy them fails because of the absence of value definiteness. Quantum mechanics “guarantees” or realizes such a scenario by demanding that any entangled quantized pair (AB) exhibits a relational encoding. The states of the individual constituents *A* and *B* are not value definite: they lack “definiteness” or “memory” or information about individual properties of its constituents—the value definiteness “resides” in the relational (not the individual), holistic, global, “collective” properties among the constituents [59]. If such individual properties are “enforced” upon the constituents through measurement, they react with a context translation which, through nesting, introduces stochasticity because of the many degrees of freedom introduced from the “outside” environment. As a result, one obtains outcome independence, although one still obtains parameter dependence; but the latter is only “recoverable” after the outcomes from both sides are compared [109,110]; locality prevails [111,112].

*Per se*, both scenarios could be extended to any type of two-partite expectation functions, which need not be linear as in the classical case, but can take on any functional form; in particular, also the quantum “stronger”-than-classical, nonlinear (trigonometric because of the projective character of the quantum probabilities) form. Indeed, by the same argument expectations and correlations might be even “stronger” than classical and quantum ones [108,113,114,115] without violating Einstein locality.

Some argue that random outcomes “save” quantum mechanics from violating relativistic causality. Because if it were possible to somehow use the relational encoding of entangled inseparable states, either by duplicating nonorthogonal states [116], or by stimulated emmission [117], then *B* could infer information on *A*’s settings even before knowing *A*’s outcome *post factum*, *posterior*, and in retrospect (after combining the knowledge of both outcomes). The random outcomes on *A*’s side assure that *B* cannot know what happens at the former side. This argument can be extended to stronger-than-quantum correlations.

However, this kind of “peaceful coexistence” [109,110] may also be seen as a characterization of the second scenario (ii) discussed earlier. In particular, if one is considering the “common share” accessible to *A* and *B*: it is, say, a pure entangled state of (AB); more formally, it is an indecomposable vector. As it is not decomposable, there is no meaning associated with individual properties of *A* and *B*. In this form, quantum entanglement defines spatio-temporal proximity, yet cannot produce any means of communication between the entangled parties: the “more entangled” the parties get, the “less individual” properties they carry. Their common share, such as indecomposable vectors, cannot give rise to any form of classical communication between the entangled parties as it is useless.

I, therefore, suggest that rather than speaking about a “peaceful coexistence” between relativity and quantum theory, we should speak of this no-signaling constraint as an unavoidable feature of emergent space-time from entanglement. The value-definiteness of the common indecomposable vector share of (AB); that is, in a value indefiniteness of the individual states of *A* and *B* results in stochasticity if individuality is forced upon those subsystems; very much in the same way as stochasticity emerges (by context translation) from coherent superpositions or linear combinations of states, when measured “along with the detuned, twisted contexts”; as sketched earlier.

## 2. Historic Perception of Randomness

In what follows, randomness will be discussed in the historic context. This is important because of the lessons one could learn for the contemporary debate and perception of lawlessness and randomness. According to an influential narrative, the European Enlightenment developed as a courageous, thorough, and highly successful—the criterion of success is taken relative to and in terms of full-spectrum dominance compared to alternative worldviews grounded in esoteric thought—exorcism of transcendence; in particular, the rejection of law-defying miracles [118]; moreover, the empirical sciences “established natural laws” of regular, reliable tempo-spacial coincidences which appear to be trustworthy and therefore of great utility.

The denial of any direct breach or “rupture” of the laws of nature ([119,120], Sect. III, 10) has pushed the boundaries of conceivable transcendental real-time interventions, and, in particular, divine providence, to the fringe of “gaps” ([119,120], Sect. III, 12) in the laws of nature— indeterminate situations where applicable laws, and thus the Principle of Sufficient Reason [121], have not (yet?) been identified.

As effective as the formal [122] and natural sciences are in terms of utility, they turn out to be as means and context relative as any construct of thought: those imaginations of the human mind cannot deliver any “Archimedean point” or “ontological anchor” upon which an “objective reality” (whatever that is) can be based.

Means relativity of an entity such as an idea or a physical theory is the dependence (eg., validity, existence) of this entity on the means, conventions, or assumptions employed. Context relativity relates to whatever are the circumstances that form the setting for an event in terms of which it can be fully understood. Perhaps means and context relativity are equivalent notions, yet the emphasis lies on different aspects of a situation.)

Indeed, it is my idealistic [123,124,125,126] observation, or rather, stance or conviction, that all our physical narratives [127,128,129], doubles [130,131], images [132,133], and—more optimistically—representations [134] of what we experience as “Nature” are metaphysical—or at least amalgamated with metaphysical components—and ultimately can be denounced as being suspended in our free thought. Therefore, historically, we experience a succession of incongruent, incommensurable [135,136,137,138,139,140] scientific research programs [141,142]; a lineup which should make us humble when it comes to the mind-boggling effectiveness [122] of some of our formalisms in predicting, programming, manipulating, and instrumentalizing physical systems. The desperation, if not nihilism, that results from the deconstruction of long-held beliefs and narratives has been very vividly described by Schopenhauer [143], as well as through Nietzsche’s *Übermensch* [144,145] and Camus’ *Sisyphe* [146].

An obvious counter-response to such idealistic positions is to contend that physics is firmly grounded in empirical data drawn from observation of experimental outcomes. Support of theoretical physical models in the form of corroboration or falsification [147,148] by empirical evidence [149] is indispensable. As an extreme demand, physical theory should strive to include only operational entities which are physically realizable in terms of achievable actions and measurements [150,151,152,153,154].

However, the history of science presents ample evidence that it has never been possible to resort to empirical evidence for the advancement or discrimination of theoretical models alone [137,141,142]. Indeed, as stated by Einstein [84] (reprinted as Letter 206 in [85], my translation), there is a metaphysical circularity because “the real difficulty lies in the fact that physics is a kind of metaphysics; Physics describes ‘reality’. However, we do not know what ‘reality’ is; we only know it through the physical description!” Furthermore, although both the prediction and the willful reproduction of phenomena appears to be the cornerstone of current natural sciences, the “empirical evidence” relating to “scientific facts” is often indirect and fragile, deserving a nuanced and careful analysis [155,156].

I shall offer three examples for the type of problems encountered in quantum mechanics; all three related to the occurrence of certain “clicks” of detectors. Arguably, the occurrence or non-occurrence of such a click is the most elementary, binary observable one could think of. However, while the (non)registration of detector clicks may be considered indisputable (for all practical purposes [93], and notwithstanding quantum erasures or haunted measurements [48,49,50,51,52,53,54,55,56]) the “meaning” of such clicks [157] remain open to a great variety of perceptions, interpretations, and understandings.

The first example is about measurements [158] of violations of classical locality with time-varying analyzers [159] if the periodic switching is synchronized with photon emmissions [160]. A second example is about a debate [161,162] on quantum teleportation [163,164]. A third example is about the contingencies [76] arising from counterfactual arguments of Hardy-type configurations [75,165]. These cases document well the different claims and aspects derived from single detector clicks, as perceived by different participating discussants.

Other aspects related to very general limits on symbolic representations need to be acknowledged. Any formalization of physical (in)determinism by (in)computability, and physical randomness as algorithmic incompressibility, and general induction [166,167,168,169,170] would require transfinite means not available [171] in this Universe [172,173,174]. This is because the associated formal proofs are blocked by the aforementioned Gödel–Turing-type incompleteness/incomputability results.

Therefore, one cannot expect that the formal and natural sciences offer absolute corroboration of any type of semantic statements. All they allow is the systematic exploitation of syntax and narratives which are true relative to the chosen means and purposes.

In what follows, we shall first discuss what general options of randomness can be imagined; and then proceed with a discussion of their concrete physical *modi operandi.*

### 2.1. Bowler Type Scenario of a Clockwork Universe

In what follows, “god” or “deity” is understood as an entity creating existence; a sort of “programmer of the Universe.” The assumption of a “clockwork universe”—that is, “stuff” such as matter, energy, together with its assorted evolution laws which are uniformly valid and unique (leaving no room for alternatives)—entails a “bowler”-type situation. The Principle of Sufficient Reason [121] rules; nothing occurs without a “reason” or “cause”. Once this universe is created *ex nihilo* and put into motion there is no further or additional interference with it; as all necessary and sufficient conditions exist to determine its evolution uniquely and completely from a “previous” state into a “later” one. In such a scenario free will appears to be illusory and subjectively, as per assumption choices are merely fictitious and delusional.

#### 2.1.1. How Could Physics Facilitate and Support Such a View?

Here are some arguments that may be put forward in favor of a bowler-type clockwork universe:(i)The description of a unique physical state as a function of some operational physical quantity such as time—indeed, the very notion of a total function (as opposed to partiality [77]), Laplace’s demon, causal [175] determinism, and the Principle of Sufficient Reason are scientific tropes and schemes signifying clockwork universes. They were widely held in pre-statistical physics and quantum areas until around *fin de siècle*.In ordinary differential equations of classical continuum mechanics and classical electrodynamics, the semantic notion of “determinism” is formalized by the uniqueness of the solutions, which are guaranteed by a Lipschitz continuity condition ([91], Chapter 17).(ii)The quantum state evolution is postulated to be unique and deterministic. Formally this is represented by a unitary transformation, that is, a generalized rotation mapping one orthonormal basis into another one. Such a state evolution is one-to-one and thus reversible and unique. However, if the preparation context differs from the measurement context, the quantum state does not identify outcomes uniquely, thereby allowing one particular kind of quantum indeterminacy. However, in general—in the case of coherent superposition or mixed states—the quantum state is not operationally accessible. Therefore this sort of quantum determinacy cannot be given any direct empirical meaning.(iii)Deterministic chaos is characterized by a unique initial value—a “seed” supposed to be taken from the mathematical continuum and thus incomputable and even random with probability one—whose information or digits are “revealed” by some suitable deterministic temporal evolution. (Idealized randomness of an infinite string is taken to be algorithmically incompressible [20].) To be suitable a temporal evolution needs to be very sensitive to changes of initial seeds such that very small fluctuations may produce very large effects. This is like Maxwell’s gap scenario discussed later.Like quantum evolution, deterministic chaos might be considered both an argument for and against classical determinism: because the assumption of the continuum renders almost all seeds formally random [20], thereby passing all statistical tests of randomness; in particular an “elementary” test such as Borel normality, certifying that all sequences of arbitrary length occur with the expected frequency, but also much stronger ones. Unfortunately, Borel normality is no guarantee of randomness because very regular sequences, for instance, the Champernowne constant [176] $C_{10}$ in base 10 is just the sequence obtained by concatenating successive numbers (encoded in base 10), turn out to be normal.In this respect, classical machinery designed to use extreme sensitivities of the temporal evolution to the initial seed, such as the Athenian [177] κληρωτηριoν (*kleroterion*), for all practical purposes is not inferior to a quantum oracle for randomness, such as *QUANTIS* [18], based on the “evangelical” belief of irreducible quantum randomness [31].(iv)In system science or virtual physics, this modus could be referred to as a very restricted virtual reality, computational gaming environment, or simulation [178,179,180,181] (*aka* simulacrum), whereby it is assumed that there is no interference from “the outside” (*aka* beyond): the respective universe is hermetic. No participation is possible; only passive (without interference) observation.

#### 2.1.2. How Could Physics Contradict Such a View?

Here are some arguments that may be put forward against a bowler-type clockwork universe:(i)Classical gaps are characterized by instabilities at singular points, such that very small fluctuations may produce very large effects. To quote Maxwell ([182], pp. 211,212), “for example, the rock loosed by frost and balanced on a singular point of the mountain-side, the little spark which kindles the great forest *…* At these points, influences whose physical magnitude is too small to be taken account of by a finite being, may produce results of the greatest importance”.(ii)In some physical situations the Lipschitz continuity is violated, yielding no unique solutions. The Norton dome [183,184] is a contemporary example of such a situation.(iii)Spontaneous symmetry breaking, a physical (re)source of non-uniqueness, is a spontaneous process by which a physical system in a symmetric state ends up in an asymmetric state. This is facilitated by some appropriate “Mexican hat” potential, not dissimilar to Norton’s dome or Maxwell’s ([182], pp. 211,212) “rock loosed by frost and balanced on a singular point” mentioned earlier.In particle physics, the Higgs mechanism, the spontaneous symmetry breaking of gauge symmetries, plays an important role in the origin of particle masses in the standard model of particle physics. All of these ruptures or breaches of uniqueness depend on the assumptions and models involved.(iv)Quantum indeterminacy, in particular, complementarity, contextuality (aka value indefiniteness), and aspects (such as the exact decay time) of the occurrence of certain single events are postulated to signify indeterminism.

Because of both formal and empirical reasons, these scenarios might not be interrelated and not separate: for instance, one might suspect that Maxwell’s instabilities at singular points could be formalized by “Mexican hat” type potentials discussed in spontaneous symmetry breaking, or by ordinary differential equations yielding Norton dome-type configurations. One might even speculate that all violations of Lipschitz continuity amount to some kind of symmetry breaking.

Empirically, one might argue that, for all practical purposes [93], Maxwell’s scenario and Norton dome-type configurations (related to violations of Lipschitz continuity) or spontaneous symmetry breaking, never “actually” happen. Because for all practical purposes a rock loosed by frost is never (with probability zero) symmetrically balanced at a singular point; rather the position of its center of gravity will fluctuate around the tip, thereby spoiling symmetry. Furthermore, one may argue that, due to (vacuum) fluctuations, singular points make no operational sense; they are (over)idealized concepts invented by the human mind for mere convenience. In particular, microscopic quantum zero-point fluctuations, and thermal fluctuations [185] ultimately spoil symmetries. Therefore, all such exploitations of such singularities might confuse epistemic convenience with an ontology that has no physical, operational grounds.

### 2.2. Scenario of a Stochastic, Disorganized Universe

The “converse” of a Laplacean determinism governed by a unique state evolution “tied to” previous states, as mentioned in the previous section, is one in which any given state is independent of the respective previous (and future) states. (Two events *A* and *B* are statistically independent if their joint probability P(A∩B) can be written as the product of their single probabilities P(A) and P(B); that is, P(A∩B)=P(A)P(B). It turns out that this results in a journey down a rabbit hole, as the concept of probability is a nontrivial one [186].) In such a most extreme scenario among many conceivable degrees of stochasticity the universe is “completely” stochastic and disorganized on the most fundamental level. For the embedded observer’s intrinsic perspective, due to irreducible contingency and chance, it appears as if such a world is constantly created anew by throwing some sort of dice.

This may be considered an extreme form of *creatio continua.* However, extrinsically—that is, from an external, extrinsic, perspective—this may be considered *creatio ex nihilo* as no active, real-time participation is assumed. Indeed, one may speculate that if the temporal ordering of events (and causality) turns out to be epistemic—an intrinsically emerging concept/observable of (self-)cognition/observation—then any differentiation based on temporal creation—such as *creatio continua* versus *ex nihilo*—turns out to be a “red herring.” Alas, without granting “time” some ontology, also differentiation between a “bowling” or “curling” god collapse.

Whether and how some sort of structural continuity of existence can emerge and be maintained under such circumstances is a fascinating question. As in such a scenario space and time, as much as notions of causality and the laws, are emergent concepts, continuity might emerge with them.

Indeed, one might speculate that “the laws” are some sort of expressions of chaos, the formation of matter and genes are expressions of these laws, the individuals carrying those genes are expressions thereof [187], and that the ideas about the world are expressions of these individuals. In that transitive way, the Universe contemplates itself through our ideas—ideas such as religion, mathematics, ethics, and so on. (This is not dissimilar to the impossible choice not to communicate [188].)

Contemporary physics supports such a view in postulating that many elementary events—such as the spontaneous or stimulated emission of photons—occur acausally, irreducibly pure, and simple [31,189]. Indeed, both classical statistical physics at finite resolution, and quantum mechanics, support such a view. (A Laplacian demon with unbounded resources might be able to determine future states from present ones with arbitrary precision.)

The Viennese *fin de siècle* physicist Exner [190,191], motivated by statistical physics and the radiation law [192], suggested that ([193], pp. 7,18) “*…* laws do not exist in nature, those are only formulated by man, he makes use of it as a linguistic and computational aid and only wants to say that the processes in nature run as if matter, like a sentient being, would obey these laws. *…* So we must understand all so-called exact laws only as average laws, which are not valid with absolute certainty, but with the higher probability the more individual processes they result from. All physical laws go back to molecular processes of random nature and from them follows the result according to the laws of probability calculus *…* .”

Even in totally “random” datasets, some sort of structure must necessarily emerge by the law of large numbers: for instance, if two dice are thrown sufficiently often, the number seven appears to be the most likely sum of their two faces. Modern arguments for the emergence of laws from chaos employ, among other methods [194,195,196,197,198,199,200,201], Ramsey theory, for structure formation and structural continuity through spurious correlations [202]. It is irrelevant whether these events occur “absolutely randomly”—indeed, as has been pointed out earlier, on an individual level and with finitistic means, “absolute randomness” appears to be a vacuous concept.

### 2.3. The Intermediate Curler Case

Intuitively, the curler case [203] is one in which the natural laws—whatever their form and origin—predominate, but there are situations in which such laws do not exist, or if laws exist, they are violated. The first “weak” case of indeterminism can be realized by gaps.

As mentioned earlier ([119,120], Sect. III, 10) “stronger” forms of curling involve a “rupture” of the laws of nature, as they are in direct violations of those laws as mentioned in Voltaire’s Philosophical Dictionary ([204], Chapter 330). Although nobody can *a priori* exclude such latter cases we shall henceforth stick with Hume’s attitude towards miracles ([156], Section X) and neglect them.

Theologically, this could be perceived as a mild form of *creatio continua* (cf. my earlier remarks on *creatio continua* in Section 2.2): god has created laws that are not violated, but god also left “some room” to communicate *via* gaps.

A “god of the gaps” has been rephrased in many ways. This concept is also quite popular since, on the one hand, the obvious regularities of experience and life express correlations or laws which appear evident: the daily cycle of the sun, the yearly cycle of the seasons, life, death; apples and other stuff falling down and not up, and so on. So denial of regularities appears futile. On the other hand, humans experience fate and uncontrollable circumstances quite often. In a similar reaction, the primitive mind (re)interpreted such “evidence” as god’s signal.

As more and more “fateful” behaviors became “understood” and even controllable—think of medical conditions and also volcanic eruptions, floods or weather phenomena such as lightning and thunder—it is not unreasonable to speculate that, maybe, eventually, there will be no such gaps left—in which case one recovers the bowler, *ex nihilo,* scenario. Alternatively some “pure” gaps in the causal fabric of our universe might “turn out”—that is, relative to the assumptions and means employed— to be irreducible and final: those gaps cannot be eliminated and might remain forever. In secular terms, this could be suspected to signify irreducible indeterminism or randomness [31]. However, there exist other, possibly transcendental, interpretations involving intentionality across gaps.

That these latter scenarios are not purely speculative can be demonstrated by an interactive gaming scenario: If one is considering an interactive virtual reality environment [205,206] one usually assumes that the virtual reality is “sustained” or “supported” by a computational process “running” on some kind of computer whose physical characteristics are not directly related to the simulacrum. To be feasible and nonmonotonic it can be assumed without loss of generality that both the universe in which the simulation is implemented and the simulated universe are capable of universal computation in the sense of Chuch–Turing. To be interactive the two universes need to be intertwined and connected by some sort of (bidirectional) gap through which information flows in both “directions”.

This could result in a sort of dialogue between those realms—a “backflow” from the simulacrum to the universe in which the simulation takes place—such that the former simulacrum performs “empirical studies” on the latter, thereby fully and actively participating in it. In this very speculative scenario, “transcendence becomes immanence.” Think of evolving artificial intelligence in a computer simulation becoming aware of its situation and asking online players questions about its situation and the general setup. However, as symmetric as an exchange through the interface may appear, it is asymmetric in one aspect: whereas the simulacrum cannot exist without the world in which the simulation takes place the latter can exist without the former.

For an intrinsic [207] observer embedded [178] in the virtual environment and bound by its operational means the capacity to send an arbitrary signal through the interface—from the simulating universe (*aka* “the beyond”) to the simulacrum—can only be realized by a gap. Because without a gap, the signal must remain immanent; that is, it reduces to either lawful or chaotic behavior.

Gaps potentially allow some “transcendental” exchange of signals but do not necessarily imply such a conversation or dialogue. Therefore, gaps are a necessary but not a sufficient condition for transcendence—just because gaps have been located does not imply the existence of “active” transcendental entities.

From a theological perspective, gaps can realize individual (human) soul/mind-body dualism [208], and also divine providence ([119,120], Sect. III, 9–16).

How does physics support gaps? Can physics rule them out? The following is an update and extension of Frank’s discussion on physical gaps.

(i)As has been mentioned earlier, in the classical domain of ordinary differential equations some breach of the Lipschitz continuity condition ([91], Chapter 17) could cause nonunique solutions. Often such types of gaps are identified with instabilities at their singular points ([182], pp. 211,212, [119,120], Sect. III, 13).(ii)As has also been discussed earlier, quantum complementarity, and, as an extension thereof, quantum contextuality (*aka* value indefiniteness) can be interpreted as the impossibility to co-represent [22,106,209] certain (even finite) sets of—necessarily counterfactual because they are complementary—quantum observables, relative to the assumptions. (One assumption entering those proofs are the (context) independence of outcomes of measurements for “intertwine” observables occurring in more than one context. For reasons of being able to intertwine contexts formalized by orthonormal bases this can only happen in vector spaces of dimension higher than two.) This is problematic as the corresponding experimental protocols (“prepare a pure state and measure a different one”) seem to suggest that they “reveal” some pre-existing property—indicated by the (non)occurrence of a detector click. This could be misleading, as the respective click might either be subject to debate and interpretation or merely signify the capacity of the measurement apparatus to “translate an improper question;” introducing stochastic noise [63]. (A debate [161,162] on the alleged “*a posteriori* teleportation” is an example for such a nonunique semantic perception of syntactically undisputed detector clicks.) This appears to be related to notorious inconsistencies in quantum physics proper [25,38,39,47,210] due to the assumption of irreversible quantum measurements.(iii)Aspects of certain individual, single events in quantized systems such as the time of emission or absorption of single quanta of light, are postulated to be indeterministic.

## 3. The (Un)known (Un)knowns

The relativity of the considerations on the respective assumptions and means invested or taken for granted results in an echo-chamber of sorts: whatever one puts in one gets out. As mentioned earlier there is no “firm (meta)physical ground,” no undisputable “Archimedean ontological anchor” upon which such speculations can be based. Furthermore, the tendency of the mind to rationalize, project [211,212,213], and empathically embrace opinions that are favorable to one’s ego-investments increases delusions about particular beliefs and corroborations thereof even further.

At this point, the reader might get frustrated: a negative message (akin to a negative theology) has been delivered. Alas, unfortunately, this is all that can be safely stated. One positive side effect might be the abandoning of what the Vienna Circle (in a Humean tradition) called “meaningless pseudo-statements” [214,215,216] targeting a particular hocus-pocus, abracadabra delusional (thought) rituals delivered by sophistic philosophers and an orthodox clergy. However, one has to be very careful not to “throw the baby out with the bathwater.” Shortly after these bold rejections of metaphysical entities, it turned out that their program based on empirical evidence and formal logic proposed could not be carried out as completely as desired [217,218,219,220,221,222,223].

Therefore, we should accept the sobering fact that there is certainty only in our uncertainty. This has been expressed by many insightful individuals of many philosophical traditions and religions and at various times. Aurelius Augustinus, for instance, writes ([224], Book XI, chapter 25.32), “Do I perhaps not know how to express what I do know? Woe is me: I do not even know what it is I do not know!”

## 4. Summary

Quantum randomness appears epistemic: identical pre- and post-selected states and observables yield definite outcomes because the vector or projection operator shares are identical. If there is a mismatch between preparation and measurement, then the measurement apparatus, as part of the environment, may “contribute” to the respective outcomes by context translation. Therefore, randomness extracted from coherent superpositions or linear combinations of the quantum state might be based on the complexity of the environment rather than on the intrinsic, ontologic “oracle” nature of the state. “Objectification”—the emergence of a property which the original state is not encoded in—is associated with this influx of information from the environment.

This readily extends into entanglement: relationaly encoded quantum shares (that can be pure entangled states represented by inseparable vectors) will not be able to render individual value definiteness of its constituents that is necessary for communication between those constituents. This relates to the concept of emergent space-time from separation through nonentanglement, and inseparability by entanglement.

In the second part of the paper, a wealth of historic resources on random physical outcomes has been reviewed. The emphasis has been on the “evangelical” side of the perception of value indefiniteness, as it has emerged historically.

## Figures and Tables

**Figure 1 entropy-23-00519-f001:**
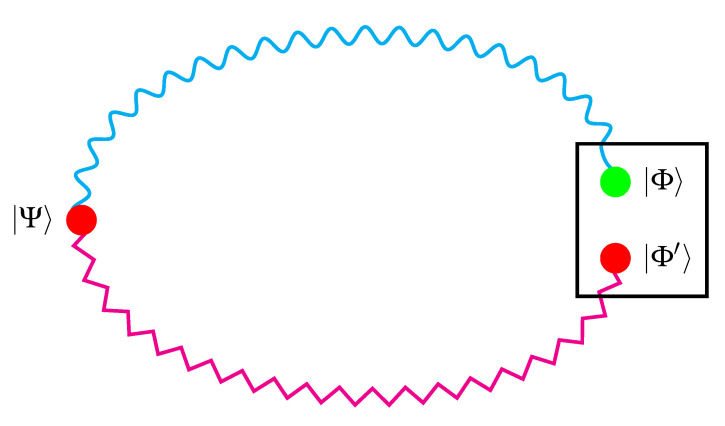
Serial composition of two gadget hypergraphs with terminal points |Ψ〉 and approaching $\left|\Phi \rangle ↔\right|\Phi^{\prime }\rangle$. The snake-like decorated curve indicates a classical true-implies-false relation. The zigzag-like decorated curve indicates a classical true-implies-true relation.

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
