# Peer review of "Quantum Randomness is Chimeric"

_entropy, 2021, doi:10.3390/e23050519_

Round 1

Reviewer 1 Report

A review of

Quantum Randomness is Chimeric

by Karl Svozil

for Entropy

This paper shows that the common belief that quantum mechanics is at its heart random is not really true. There are two types of randomness: epistemological and ontological. Epistemological randomness means that we simply do not know what the results of an experiment will be. This has to do with the lack of knowledge that human beings possess. In contrast, ontological randomness says that the problem is not us. Rather, the system is random or chaotic by itself. It is not that we cannot predict the results of the experiment; it is that the results cannot be known even theoretically. The author’s point is that epistemological randomness pervades our universe so much, that we can never have a system that is purely ontologically random.  Another way to say this is that whether or not a system is ontologically random is metaphysical and unanswerable. He proves his point with many examples.

The writer is a well-known physicist who has written much on the foundations of quantum mechanics and in many other areas. His opinions are respected and his insights are deep. While many might not agree with him on these points, it will generate discussion and it is a nice contribution to the literature.  

The paper closes with a wide-ranging historical discussion of determinism and randomness. This is combined with an extensive bibliography with 212 entries. The author clearly knows the literature and presents his case well.

I did not find any typos or errors in the paper.

I strongly urge Entropy to publish this important paper.   

Author Response

Thank you for this nice and kind review.

Reviewer 2 Report

This is not a scientific article, so I cannot recommend it for publication

in Entropy.

Author Response

There is a certain variance in the perception of what constitutes a scientific article. This is a longstanding debate in the philosophy of science. My stance is a tolerant attitude while maintaining scientific standards.

As all of this is not objectifiable this is not a straightforward matter.

Reviewer 3 Report

I would like congratulate you for the work, the idea is novel and I think the result you are reporting is based on your understanding of amount of reference.  The manuscript is written using an elegant stile. It looks like reasonable but I can rarely confirm it, even though I hope it can be published so the society could argue on your conclusion.

Author Response

Thank you for these kind and respectful, encouraging words!

Reviewer 4 Report

The paper is interesting and worth publishing, but I hardly support the acceptance  as a contribution to “Entropy”. The paper has a clearly foundational character, nicely analyses some aspects of randomness in quantum mechanics (and of randomness in general), but does not bring particular results progressing our state of knowledge. That is why I propose to submit it to a journal publishing more fundamental analyses (e.g. ”Foundations of Physics” or “Foundations of Science”).

The paper consists of three parts, each concerning some fundamental problem of quantum indeterminism. The topics are intertwined, but the author failed to show it convincingly. In the first part (Sections 1.1 and 1.2) the  author (in his words) “discusses randomness `extracted’   from measurements 3 of coherent superpositions of classically mutually exclusive states”. The discussion is interesting, but I am not sure what are the conclusions of the authors reasoning. It seems that, apart from introducing anew nice terms to the discourse (“chimeric” and “objectivization”), one does not go beyond (again, in authors words) the following "consequences for the stochasticity of chimeras: they are not only based on some property intrinsic to the object but on the combined context by which the object, as well as the apparatus, is defined”. Is this something more than the commonly accepted opinion that stochasticity is introduced to quantum mechanics by measurements?

The second part (the rest of Section 1) is devoted to the idea that phase-space is not, as commonly meant, a kind of scaffolding for physical theories, but rather an emergent phenomenon. The idea of the emergence of space-time structures was recently put forward in investigations of relations between gravity and quantum mechanics. The novelty of the approach proposed in the paper is that correlations (entanglement) is the source of emergence. Alas, the paper contains only a very rough outline of a research program concerning this problem, no new results are reported.

The third part is a historical outline (with clear philosophical overtones) of the approach to stochasticity (randomness, indeterminacy) in physics. It discusses the concept of the fully deterministic, fully chaotic and, in a certain sense, an intermediate model of the universe. I am also not sure what are the physical conclusions. Moreover, some arguments of the author are, in my opinion, misguiding, although, admittedly, they are not of the highest importance. For example, in his discussion of the “clockwork universe”, the author invokes several times non-uniqueness of solutions of dynamical equations (caused by violation of Lipschitz conditions, usually assumed to assure uniqueness), and the so-called “Norton dome” as an example of such a situation. It seems to me that the fault is on the mathematical description and not on the physical side. We probably use not adequate mathematical instruments to describe a classical point-particle at the exact top of the dome of a certain shape. Such a system is non-physical, and the fact that the instruments are sufficient to explain the motion of finite-size balls rolling e.g. on the hemisphere, does not give any hints that it is equally appropriate for a point-particle moving on the Norton dome.

Concluding, I reiterate:  the paper is interesting and can be a staring point to further discussions. As such it is worth publishing, but definitely not in “Entropy”.  

Author Response

There are three main criticisms:

(i) the paper should be published in some more "foundational journal, not in Entropy;

(ii) the paper re-iterates a well-known fact: that measurement introduces stochasticity;

(iii) that many arguments for classical indeterminism are based upon superficial idealizations.

Wrt (i) I would like to point out that, as I have noted, there exist strong resemblance between the debate on quantum measurements and the (historic) debate in statistical physics of how exactly irreversibility (and the second law of thermodynamics) could emerge from microphysical determinism. I truly believe that, therefore, Entropy is just the right journal for publishing these considerations.

Wrt (ii) my point of departure from the "common" view is the "nesting" aspect of the situation, as outlined already by Everett and Wigner; but unlike them, more in the spirit of statistical physics: to state it in Maxwell's words: the stochastic behavior (and entropy increase) originates from sampling; rather than taking this for granted.

I have therefore added a paragraph expressing this:

"One might object that this stance reiterates a well-known fact: that quantum measurement introduce stochasticity. The point of departure from this common view is the emphasis on the ``nesting'' aspect of the situation, as outlined already by Everett~\cite{everett} and Wigner~\cite{wigner:mb}; but unlike them, more in the spirit of statistical physics: in a Maxwellian  view~\cite{Myrvold2011237} the stochastic behavior (and entropy increase) originates from sampling---from not looking at the micro-physical level but at some ``aggregates''---rather than taking this for granted. "

Wrt (iii) I agree with the Referee. A paragraph of the paper mentions this, beginning with

"Empirically one might argue that, for all practical purposes~\cite{bell-a}, Maxwell's scenario and Norton dome-type configurations (related to violations of Lipschitz continuity) or spontaneous symmetry breaking, never ``actually'' happen.  ..."

Round 2

Reviewer 4 Report

As I have already written in my first report I found the paper interesting and worth publication. Some of the author's explanations and additions strengthened my positive opinions. But the main issue was whether "Entropy" is a proper choice for publication of the paper. I still have doubts, but leave the decision to the Editors.